# A Novel Energy-Efficient Contention-Based MAC Protocol Used for OA-UWSN

**DOI:** 10.3390/s19010183

**Published:** 2019-01-07

**Authors:** Jingjing Wang, Jie Shen, Wei Shi, Gang Qiao, Shaoen Wu, Xinjie Wang

**Affiliations:** 1School of Information Science and Technology, Qingdao University of Science & Technology, Qingdao 266061, China; wangjingjing@qust.edu.cn (J.W.); 4016110010@mails.qust.edu.cn (J.S.); 2College of Underwater Acoustic Engineering, Harbin Engineering University, Harbin 150001, China; qiaogang@hrbeu.edu.cn; 3Department of Computer Science, Ball State University, Muncie, IN 47304, USA; Swu@bsu.edu; 4School of Information and Control Engineering, Qingdao University of Technology, Qingdao 266000, China

**Keywords:** MAC protocol, OA-UWSN, high throughput, low energy, contention-based

## Abstract

A hybrid optical-acoustic underwater wireless sensor network (OA-UWSN) was proposed to solve the problem of high-speed transmission of real-time video and images in marine information detection. This paper proposes a novel energy-efficient contention-based media access control (MAC) protocol (OA-CMAC) for the OA-UWSN. Based on optical-acoustic fusion technology, our proposed OA-CMAC combines the postponed access mechanism in carrier sense multiple access with collision avoidance (CSMA/CA) and multiplexing-based spatial division multiple access (SDMA) technology to achieve high-speed and real-time data transmission. The protocol first performs an acoustic handshake to obtain the location information of a transceiver node, ensuring that the channel is idle. Otherwise, it performs postponed access and waits for the next time slot to contend for the channel again. Then, an optical handshake is performed to detect whether the channel condition satisfies the optical transmission, and beam alignment is performed at the same time. Finally, the nodes transmit data using optical communication. If the channel conditions do not meet the requirements for optical communication, a small amount of data with high priority is transmitted through acoustic communication. An evaluation of the proposed MAC protocol was performed with OMNeT++ simulations. The results showed that when the optical handshaking success ratio was greater than 50%, compared to the O-A handshake protocol in the literature, our protocol could result in doubled throughput. Due to the low energy consumption of optical communication, the node’s lifetime is 30% longer than that of pure acoustic communication, greatly reducing the network operation cost. Therefore, it is suitable for large-scale underwater sensor networks with high loads.

## 1. Introduction

Underwater wireless sensor networks (UWSN) are typically used to monitor underwater environments and collect underwater data, such as marine data collection, pollution monitoring, marine exploration, assisted navigation, and tactical monitoring [1,2,3,4]. The vast areas to be detected in marine environments result in sparse deployment of the network and widespread use of mobile sensors. In order to cover the entire target area, the size of a network is generally large, and the topology is usually based on multi-hop wireless [5]. 

Three ways are normally employed to transfer data between UWSN nodes, namely, underwater electromagnetic communication, underwater acoustic communication, and underwater optical communication. Electromagnetic waves attenuate very quickly in water, so they are rarely used [6]. Underwater acoustic communication is the most mature technique, because acoustic signal attenuation underwater is small and thus can achieve a relatively long-distance transmission (usually used for UWSN data transmission), but the bandwidth of acoustic communication is narrow and the delay is long, which is not suitable for large-capacity data transmission such as images and video [7]. Underwater optical communication has the advantages of high speed and low power consumption [8], and the attenuation of underwater blue-green light is relatively low, enabling underwater short-distance and large-capacity data transmission [9]. In recent years, combining the advantages of acoustic communication and optical communication, a new underwater data transmission mode has been proposed [10,11]. The routing algorithm of the design in Reference [12] improved the lifetime of underwater acoustic communication nodes. Compared to acoustic communication, optical communication has inherent advantages in node lifetime. Reference [13] proposed a solution for the combined use of acoustic communication and optical communication that addresses the bandwidth limitation of the acoustic channel while achieving optical alignment through acoustic position, thereby performing optical communication. Reference [14] proposed that autonomous underwater vehicles (AUVs) use long-distance acoustic communications, nodes using high-bandwidth optical communications, and a data transmission mode combining two communication modes. Compared with references [13,14], we previously proposed and designed a new underwater networking method: A hybrid optical-acoustic underwater wireless sensor network (OA-UWSN) [15]. Using optical communication for high-speed and short-distance data transmission, and using acoustic communication for control commands and node positioning, the OA-UWSN enables solving the problem of high-speed and low-cost wireless transmission of real-time or quasi-real-time video and images in marine information detection, while keeping system power consumption small.

In a wireless sensor network, a media access control (MAC) protocol is critical, and is used to prevent collisions when more than two transmissions occur [16]. At present, the network layer routing protocol has only been tested in simulations, such as the multi-level routing protocol for acoustic-optical hybrid underwater wireless sensor networks (MURAO) routing protocol [17]. The existing underwater wireless sensor network MAC protocols are only for a single transmission medium network, and there is no mature MAC protocol applicable to an underwater optical-acoustic hybrid wireless sensor network.

At present, MAC protocols adopted by UWSNs based on underwater acoustic communication have been studied for years, generally falling into two types: A MAC protocol based on time slot allocation and a MAC protocol based on contention. References [18,19] proposed that one node be responsible for scheduling the allocation of time slots across the entire network. Meanwhile, Reference [18] proposed a scheme based on time division multiple access (TDMA), where the sink node detects its distance from all neighboring nodes and notifies all nodes of the scheduling transmission sequence through superframes. Reference [19] proposed a slotted floor acquisition multiple access (S-FAMA) protocol, which divides time into slots. All packets must be transmitted at the beginning of each time slot, so that collisions can be avoided. In a contention-based MAC protocol, all nodes equally access the channel through competition. Reference [20] proposed that in an ordered carrier sense multiple access (CSMA), the scheduler notifies all nodes of the transmission schedule so that each node knows when to transmit information. Reference [21] proposed a low-energy MAC protocol (T-Lohi), which contends for the channel by sending a short frame. If there is only one node competing, then the data can be transmitted. Otherwise the competing node waits and then competes again. At present, underwater optical communication generally uses blue-green light in the visible light band for data transmission. However, UWSN research based on underwater optical communication is full of challenges. In particular, there is great need for an efficient MAC protocol.

This paper proposes a novel optical-acoustic competitive MAC protocol (OA-CMAC) based on the underwater optical-acoustic hybrid wireless sensor network in our previous work. The protocol combines the advantages of fast transmission, low loss, high bandwidth of underwater optical communication, and long-distance propagation of underwater acoustic communication [11]. Based on optical-acoustic fusion technology, it uses the postponed access of carrier sense multiple access with collision avoidance (CSMA/CA) technology to achieve high-speed and reliable transmission of underwater data while ensuring channel utilization and energy efficiency.

## 2. Underwater Optical-Acoustic Hybrid Wireless Sensor Network Topology and Data Transmission Process

As shown in Figure 1 [15], the network topology of the OA-UWSN is composed of fixed nodes, mobile nodes, sink nodes, and control centers. The fixed nodes are deployed on the sea floor to collect monitoring data (such as temperature, salinity, and depth), images, or videos. The mobile nodes are suspended in the sea, completing the data collection and passing the data to the upper mobile nodes or the sink nodes and then to the control centers. Sink nodes and control centers are deployed in the sea and receive the collected information and send it to remote terminals.

In the underwater optical-acoustic hybrid wireless sensor network, data communication is mainly divided into three phases. In the first phase, an acoustic handshake is performed to obtain the position information of a transceiver node to ensure that the channel is idle. Otherwise, a postponed access is performed to wait for the next time slot to compete again. In the second phase, according to the position information of the transmitting and receiving nodes obtained in the first stage, the node performs beam alignment (that is, optical handshake) to detect whether the signal-to-noise ratio or the bit error rate threshold required for optical communication is satisfied. In the third phase, if the optical communication requirement is met, data are transmitted between nodes through optical communication. If optical communication requirements are not met, data are transmitted between nodes through acoustic communication.

## 3. OA-UWSN MAC Protocol Design

### 3.1. Data Link Layer Channel Access

In this paper, combined with the postponed access mechanism in the CSMA/CA data link layer access method, a spatial multiplexing-based spatial division multiple access is introduced, making full use of the directional transmission characteristics of light to increase the data transmission distance. Figure 2 shows a superframe structure designed for the OA-UWSN in this paper. The protocol divides a superframe into different periods: An acoustic handshake period, an optical handshake period, a data transmission period, and a postponed access period.

### 3.2. Data Transmission, Reception, and Confirmation

This paper combines the advantages of both optical and acoustic communication modes and proposes an OA-CMAC communication mechanism. How node A and node B of the lower layers transmit data to the upper node is illustrated as an example to describe the data sending, receiving, and confirming mechanisms. As shown in Figure 3, the specific steps are as follows:The mobile node is bound to a pressure sensor, which can measure its depth and send an acoustic interrogation signal at a certain frequency. This signal can reach the water surface directly at Time 0, or it can be forwarded by the fixed node and reach the water surface at Time 1. The time difference and the measured depth are used to locate the mobile node [22];The lower nodes A and B need to send acoustic RTS1 and RTS2, respectively, to perform an acoustic handshake. The acoustic RTS (Request to Send) includes information such as location so that the next optical handshake can be completed more smoothly;After receiving the first RTS1, the upper node immediately replies with an acoustic CTS (Clear to Send). If all the nodes receive the CTS, they can know which nodes are competing to the channel, and the uncommitted nodes perform postponed access and wait for the channel to be idle;After successful competition, the lower node A sends an optical RTS to perform an optical handshake. After receiving the RTS, the upper node responds with an optical CTS to confirm whether the communication mode is optical or acoustic;If the optical handshaking is not completed within the time slot, the node that competes in the channel through the acoustic handshake performs acoustic communication with the upper node. If the two instances of handshaking are successful, optical data communication is performed. At this time, the upper node broadcasts a busy-tone signal through the acoustic signal to notify all lower nodes that it is busy, and the node that succeeded in competition performs optical communication according to the communication mode, modulation, and coding information determined by the optical handshaking;After the end of one frame of data transmission, the upper node first sends an ACK (Acknowledgement) confirmation signal to the lower node. Then the upper node broadcasts a free signal through the acoustic signal, and all lower nodes recompete for the next frame of information transmission.

Each frame is subjected to an acoustic and optical handshake, and the position information of the mutual communication node is transmitted during the acoustic handshake. If the node moves and the position information changes, the optical alignment is realigned in the optical handshake phase.

### 3.3. Analysis of the Upper Limit of the Optical Handshake Time

In a frame of the superframe structure, the acoustic handshake time is fixed. In contrast, the optical handshaking needs to perform optical alignment through beam training. Therefore, the time required for the optical handshaking is uncertain. If it lasts too long, the throughput of the OA-UWSN is not necessarily higher than that of a UWSN with pure acoustic communication. Therefore, it is necessary to limit the optical handshake time to obtain throughput that is superior to pure acoustic communication. This section presents the relationship between the optical handshaking success rate and the upper limit of the optical handshake time when the optical communication rate and the acoustic communication rate are fixed.

Network throughput refers to the ratio of all received packet bytes to the simulation time during the simulation time, as shown in Equation (1):(1)$$\mathrm{Throughput}=\frac{\mathrm{received}\;\mathrm{packet}\;\mathrm{bytes}}{\mathrm{simulation}\;\mathrm{time}}.$$

In the OA-UWSN, with the denotation that the frame length is *n*, the acoustic handshake time is *c*, the optical handshake time is *d*, the acoustic bit rate is *x*, the optical bit rate is *y*, the success rate of the acoustic handshake is β, the optical handshake success rate is α, the simulation time is *t*, and then the throughput expression of the OA-UWSN is shown in Equation (2):(2)$$T_{1}=\frac{\left(1-\alpha \right)\beta \left(n-c-d\right)x+\alpha \beta \left(n-c-d\right)y}{t}.$$

The network throughput when purely using acoustic communication is
(3)$$T_{2}=\frac{\beta \left(n-c\right)x}{t}.$$

If the OA-UWSN throughput is higher than the pure acoustic network throughput, namely $T_{1}>T_{2}$, then
(4)$$d<\frac{\alpha \left(n-c\right)\left(y-x\right)}{x+\alpha \left(y-x\right)}.$$

The upper limit of the optical handshaking time can be obtained by deriving Equation (4). Table 1 lists the parameter settings for the simulation in Equation (4). Figure 4 shows the relationship between the upper limit of the optical handshake time and the optical handshake success rate.

Figure 4 shows that as the success rate of the optical handshaking increases, the upper limit of the optical handshake time gradually increases. If the optical handshaking success rate is constant, the throughput of the OA-UWSN is higher than that of a pure acoustic communication network only when the optical handshake time is lower than the upper limit. For example, if the optical handshake success rate is 20%, the throughput of the OA-UWSN is higher than that of a pure acoustic UWSN only when the optical handshake time is less than 1.4 s.

## 4. OA-UWSN MAC Protocol Simulation Environment Settings

### 4.1. Optical Properties of Underwater Channels

Water is a complex physical, chemical, and biological system. The optical properties of water are related to the three main factors: Pure water, dissolved substances, and suspension [23]. The effects of various components on the beam in a water medium lie in their absorption and scattering effects on the beam [24].

#### 4.1.1. Attenuation Effect of Water

The incident light can be absorbed or scattered in water. The total effect of these two mechanisms is called attenuation, resulting from water molecules and atoms, biomass, and chemicals dissolved in seawater. Underwater, blue-green light attenuation at a wavelength of 450–530 nm is much smaller than other light wavebands, so blue-green light [25,26] is generally assumed for underwater optical communication.

The total absorption coefficient of water can be expressed as the sum of the absorption coefficients of various substances in water, as shown in Equation (5):(5)$$a\left(\lambda \right)=a_{w}\left(\lambda \right)+a_{ph}\left(\lambda \right)+a_{d}\left(\lambda \right)+a_{y}\left(\lambda \right).$$

In Equation (5), $a_{w}$ is the absorption coefficient of pure water, $a_{ph}$ is the absorption coefficient of chlorophyll or phytoplankton, $a_{d}$ is the absorption coefficient of nonpigment suspended particles, and $a_{y}$ is the absorption coefficient of yellow substances.

When light travels in water, the scattering effect causes the light to deviate from the original propagation direction. The scattering of light by water is caused by water itself, chlorophyll, and suspended particles [27]. The scattering coefficient is shown in Equation (6):(6)$$b\left(\lambda \right)=b_{w}\left(\lambda \right)+b_{ph}\left(\lambda \right)+b_{d}\left(\lambda \right).$$

In Equation (6), $b_{w}$ is the scattering coefficient of pure water, $b_{ph}$ is the scattering coefficient of chlorophyll or phytoplankton in water, and $b_{d}$ is the scattering coefficient of nonpigmented suspended particles. The parameters in Equations (5) and (6) are all relative to the wavelength $m^{-1}$ in units.

#### 4.1.2. The Model of Energy Attenuation

The loss of light waves during their propagation is shown in Equation (7) [28]:(7)c(λ)=a(λ)+b(λ).

In Equation (7), a(λ) is the total absorption coefficient, b(λ) is the total scattering coefficient, and λ is the wavelength.

The attenuation law of energy transmitted by light in water is exponentially distributed [29], and the average power received by the receiver is shown in Equation (8):(8)$$P_{r}=P_{t}exp[-c\left(\lambda \right)r].$$

In Equation (8), $P_{t}$ is the transmitted optical power, r is the underwater communication distance of the optical signal, and c(λ) is the total attenuation coefficient of the light beam propagating in seawater.

Considering the loss of the system device and optical path expansion, the received power of underwater wireless optical communication is shown in Equation (9) [30,31]:(9)$$P_{r}=P_{t}\eta \eta_{t}\eta_{r}\left[1-exp(-\frac{2\partial_{r}^{2}}{\theta^{2}R^{2}}\right)]exp[-c\left(\lambda \right)r].$$

In Equation (9), $\eta_{t}$ is the light-emitting power generated by the emitting device represented, $\eta_{r}$ is the light source representing the received power generated by the receiving device of the light source, η represents the noise power, and $\partial_{r}$ is the aperture radius of the optical receiving antenna. The parameter value settings are shown in Table 2.

### 4.2. Acoustic Properties of Underwater Channels

Marine media are very complex sound propagation channels that are subject to various natural conditions, geographical conditions, and random factors, which makes their physical properties very complex and unstable and causes delays, distortions, loss, and other changes in the transmission of underwater acoustic signals, loss, and other changes, which become the key factors affecting the transmission of sound waves underwater [32].

#### 4.2.1. Propagation Loss

A seawater medium is a non-ideal loss medium. During the propagation of acoustic waves, the intensity of the seawater is attenuated in the propagation direction. Propagation loss is often used to represent the loss of acoustic signal energy in the ocean [32]. It can be assumed that the propagation loss consists of two parts: Expansion loss and attenuation loss. The extended loss is the geometric effect where the sound intensity is regularly weakened when the acoustic signal propagates outward from the sound source. It is also called geometric loss. Attenuation losses include absorption and scattering, where absorption refers to the effect where part of the acoustic energy is lost due to conversion to thermal energy.

The overall acoustic path loss can be estimated using the semi-empirical formula shown in Equation (10) [33]:(10)$$A\left(l,f\right)={\left(l/l_{r}\right)}^{k}a{\left(f\right)}^{l-l_{r}}.$$

In Equation (10), ${\left(l/l_{r}\right)}^{k}$ is the expansion loss, which is related to the propagation mode and distance of sound waves; *l* indicates the distance traveled; *l_r_* indicates the reference distance; *k* is the expansion loss index, according to different acoustic propagation conditions, whose value varies usually between 1 and 2, corresponding to cylinder expansion and spherical expansion; and $a{\left(f\right)}^{l-l_{r}}$ is attenuation loss, where *f* is the signal frequency and a(f) is the absorption coefficient, which increases significantly with an increasing frequency of sound waves, in addition to temperature and depth.

#### 4.2.2. The Model of Energy Attenuation

The energy consumption model [34] for acoustic communication in underwater nodes is shown in Equation (11):(11)$$E{\left(r\right)}_{}=P{•}_{}T_{p}•A\left(r\right).$$

In Equation (11), $E_{}\left(r\right)$ is the energy required for the node to send information, $T_{p}$ is the data transmission time, *P* is the lowest power that each node can normally receive a packet, and A(r) is the energy attenuation of the data packet when the underwater transmission distance is *r*. Equation (12) is
(12)$$A\left(r\right)=r^{k}a^{r}.$$

In Equation (12), *a* is the attenuation coefficient, r is the communication distance between nodes, *k* is the energy diffusion factor and is generally 1.5, and $a=10^{a\left(f\right)/10}$ is determined by the absorption coefficient a(f) [35]. Equation (13) is
(13)$$a\left(f\right)=0.11\frac{10^{-3}f^{2}}{1+f^{2}}\;+\;44\frac{10^{-3}f^{2}}{4100+f^{2}}\;+\;2.75\;\times \;10^{-7}f^{2}+3\;\times \;10^{-6}.$$

In Equation (13), *f* is the carrier frequency, the unit is kHz, and the absorption coefficient unit is dB/m.

The parameter settings are shown in Table 3.

## 5. Simulation Results

### 5.1. Simulation Parameters

The performance evaluation of the proposed MAC protocol of the OA-UWSN was conducted on the OMNeT++ platform. The OA-CMAC protocol was compared to the OA-MAC [36] protocol based on the OA-UWSN proposed by our research group. The performance was evaluated on metrics including throughput, energy consumption, and other indicators evaluating protocol performance.

The settings of the system simulation parameters are shown in Table 4.

### 5.2. Simulation Results

#### 5.2.1. OA-CMAC Protocol Throughput

According to the difference in the optical handshaking success rate, this work simulated the relationship between the packet sending rate and the throughput of the OA-CMAC protocol and compared it to the OA-MAC protocols, as shown in Figure 5, Figure 6 and Figure 7.

In Figure 5, when the optical handshaking success rate was 20% and the network load was low, the OA-CMAC protocol had a postponed access window and the time slot utilization was low, so the throughput was lower than the OA-MAC. As the load increased, the throughput of the OA-MAC protocol based on node slot allocation tended to be stable, while the OA-CMAC protocol node randomly competed, the slot utilization rate increased, and the throughput doubled.

In Figure 6, when the optical handshaking success rate was 50%, the data packet transmission rate was low. When the network load was low, the OA-CMAC protocol had a postponed access window, and the time slot utilization was low, the OA-MAC throughput based on time slot allocation had greater advantages. With increases in the data packet transmission rate and the network load, the OA-MAC protocol data conflict rate was higher. However, the OA-CMAC protocol adopted a postponed access mechanism, which effectively avoided collisions and had a high channel utilization rate.

In Figure 7, where the optical handshaking success rate was 80%, even if the packet transmission rate was low, the throughput of the OA-CMAC protocol was higher than the other two protocols because optical communication could perform high-speed and large-capacity data transmission.

The optical handshaking success rates were configured as 20%, 50%, and 80%. When four packets were transmitted every 10 s, the throughput was 75 kbit/s, 100 kbit/s, and 150 kbit/s, respectively. With the increasing success rate of optical handshaking, the amount of data transmitted from optical communications was continuously increasing, and the throughput of the OA-UWSN was continuously improving.

The simulation results showed that, with the increase in the optical handshaking success rate, the throughput of the OA-CMAC protocol was multiplied. Even if the optical handshaking success rate was low, when the network load was high, the throughput of the OA-CMAC protocol was higher than the other two protocols. Therefore, the OA-CMAC protocol is suitable for high-speed and high-load networks.

In Figure 8, we compared the total amount of data from the OA-CMAC to the optical-acoustic simultaneous handshake MAC protocol (OAs-CMAC), which exchanges RTS/CTS messages over both acoustics and optics at the same time. Figure 8 shows the average results of 1000 simulations.

Optical and acoustic simultaneous handshakes were performed within 0.4 s in OAs-CMAC. If the optical handshake was successful within 0.4 s, optical communication was performed. Otherwise, acoustic communication was performed, and at the same time, the optical handshake continued. If the optical handshake was successful within the next 1 s, the optical communication was switched. Otherwise, the acoustic communication was performed continuously.

Since optical transmission and reception requires directional optical communication, it was difficult for RTS/CTS to perform optical alignment without obtaining the position information of the node, and the optical RTS/CTS signal was likely to be unreceived, which made the handshake meaningless. However, if an omnidirectional optical handshake was used, the handshake distance was very short, so it was also possible that the RTS/CTS signal of the optical handshake was not received, and acoustic communication was forced to be used. Therefore, the total amount of data transmitted at the same time was much lower than that of OA-CMAC. Therefore, the first acoustic handshake was used, and the location information of the transmitting and receiving nodes was transmitted simultaneously during the handshake, and then the optical RTS/CTS was performed by positional information obtained by acoustic handshake.

#### 5.2.2. OA-CMAC Protocol Node Energy Consumption Simulation

In this paper, the initial energy of the OA-UWSN node was set to 3600 kJ, and the energy threshold of the node was 1080 kJ. When the node energy was lower than 360 kJ, the node was considered dead, and the control center replaced the new node. When the node energy was 360 kJ–1080 kJ, the node stopped acoustic communication and preferentially used optical communication to transmit small-capacity data with high priority, such as temperature, salinity, and depth. If the optical handshake was not successful, the data was sent via acoustic communication.

Since the OA-CMAC protocol proposed in this paper increases the limitation of the energy threshold, the lifetime of the UWSN node is further extended compared to the OA-CMAC that does not set that energy threshold. Figure 9 shows node energy consumption simulations in the OA-CMAC and OAs-CMAC protocols. The simulation curve is the node’s two lifetimes. In each curve, the gentler part of the decrease is the energy consumption during the acoustic handshake and optical handshake. The steep part is the energy consumption of optical communications, and the steepest is the loss of acoustic communication energy. As can be seen from the figure, the lifetime of the OA-UWSN node was 7800 h, the lifetime of the OA-UWSN node without the energy threshold was 6400 h, and the lifetime of a single acoustic communication node was 6000 h. Therefore, the lifetime of the OA-UWSN node increased by 14% compared to the OA-UWSN node without an energy threshold, which was 30% longer than the lifetime of the OAs-CMAC.

## 6. Conclusions

This paper presented a low-power contention-based MAC protocol for the OA-UWSN. The OA-CMAC protocol locates the nodes through acoustic handshaking, performs beam alignment through optical handshaking, and transmits data after successful acoustic and optical handshakes. Through OMNeT++ simulation, it could be seen that when the optical handshaking success ratio was higher than 50%, the throughput was increased by several folds compared to the underwater acoustic communication MAC protocol. The node’s lifetime was increased by 30%, greatly reducing the network cost. Using the OA-CMAC protocol can achieve higher throughput and lower node energy consumption, which is suitable for underwater wireless communication environments with high loads and large data.

## Figures and Tables

**Figure 1 sensors-19-00183-f001:**
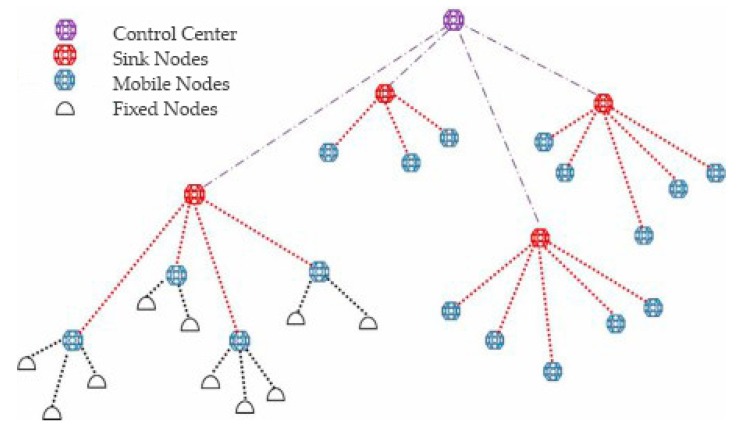
Network topology.

**Figure 2 sensors-19-00183-f002:**
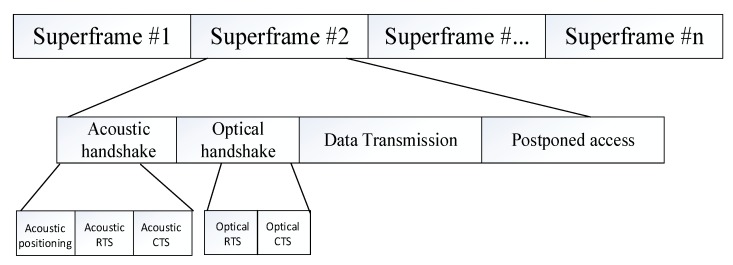
Superframe structure.

**Figure 3 sensors-19-00183-f003:**
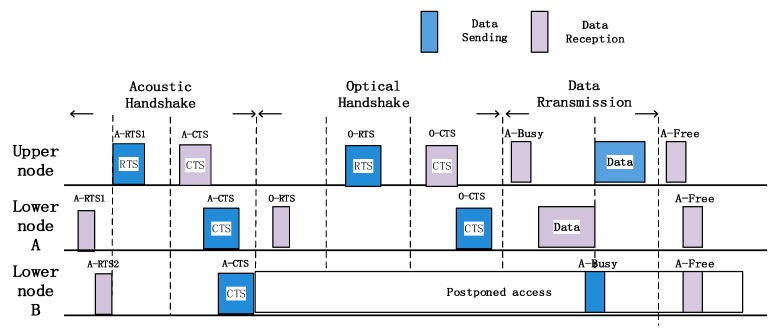
Data transmission between nodes.

**Figure 4 sensors-19-00183-f004:**
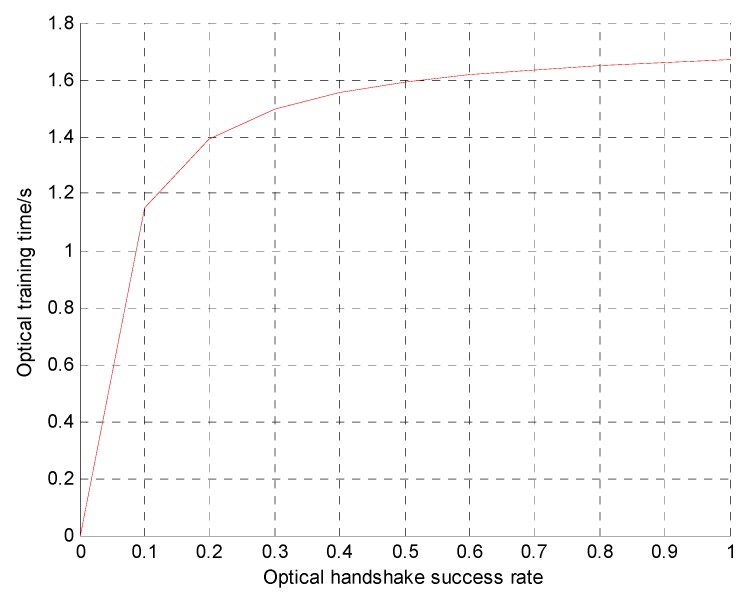
Relationship between optical handshake success rate and optical handshake time.

**Figure 5 sensors-19-00183-f005:**
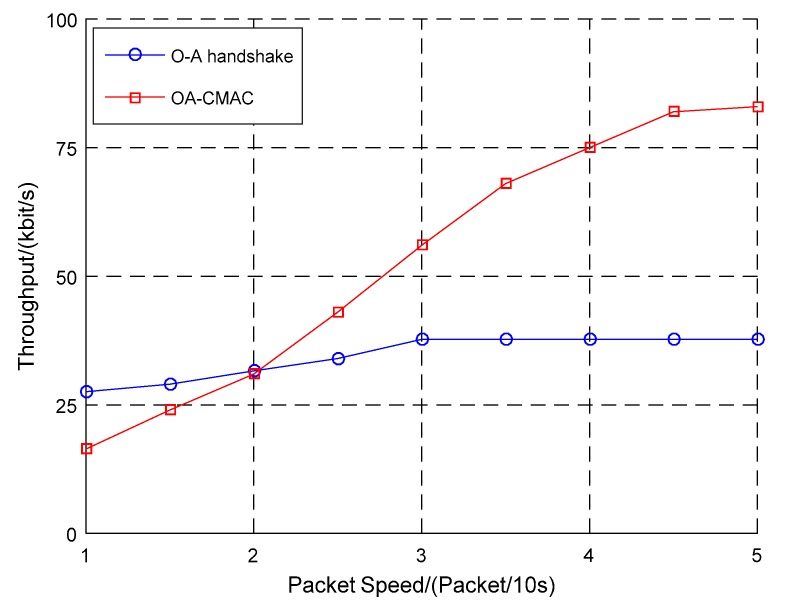
Comparison of three protocol throughputs when the optical handshaking success rate was 20%.

**Figure 6 sensors-19-00183-f006:**
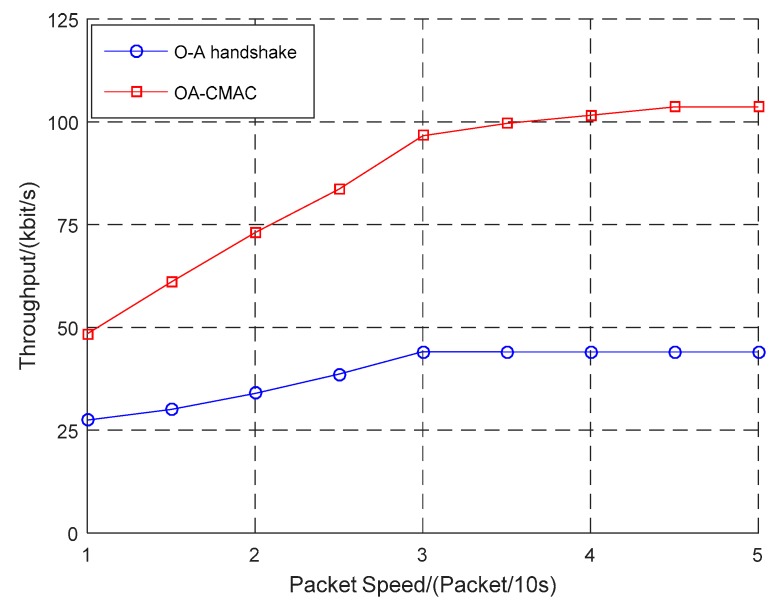
Comparison of three protocol throughputs when the optical handshaking success rate was 50%.

**Figure 7 sensors-19-00183-f007:**
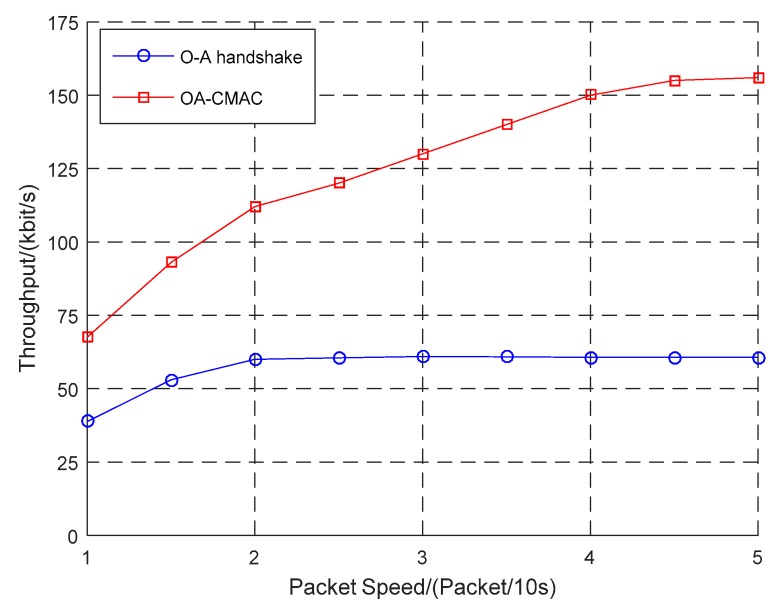
Comparison of three protocol throughputs when the optical handshaking success rate was 80%.

**Figure 8 sensors-19-00183-f008:**
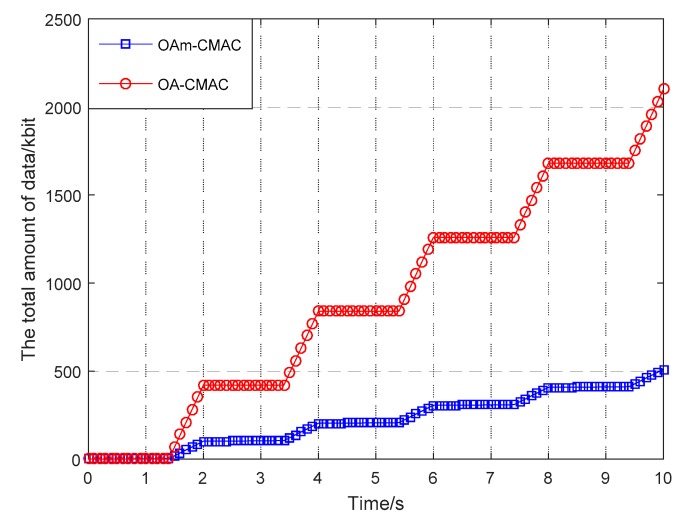
Comparison of the two protocols’ total data volume.

**Figure 9 sensors-19-00183-f009:**
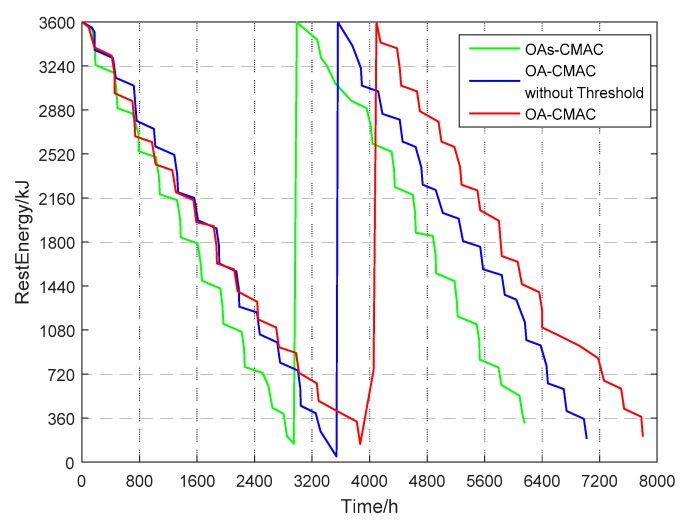
Optical-acoustic energy-efficient competitive media access control (OA-CMAC) protocol node energy consumption simulation.

**Table 1 sensors-19-00183-t001:** Channel simulation parameters.

Parameters	Value
Frame length	2 s
Acoustic handshaking time	0.4 s
Optical handshaking time	d
Acoustic bit rate	10 kbps
Optical bit rate	1 Mbps

**Table 2 sensors-19-00183-t002:** Optical energy consumption parameters.

Parameters	$\eta_{t}$	$\eta_{r}$	Caliber D (mm)	Divergence Angle θ (mrad)	Sensitivity (μW)	Attenuation Coefficient $\left(m^{-1}\right)$
Value	0.91	0.91	6	1.35	1	1.5371

**Table 3 sensors-19-00183-t003:** Acoustic energy consumption parameters.

Parameters	*P*	*F*	*R*	*k*
Value	3 mW	25 kHz	20 m–36 m	1.5

**Table 4 sensors-19-00183-t004:** System simulation parameters.

Parameters	Value
Fixed node	8
Mobile node	4
Sink node	2
Control center	1
Topological area	500 m × 500 m
Distance between fixed nodes	30 m
Distance between fixed node and mobile node	20 m–36 m
Simulation time	8000 h
Number of simulations	1000
Initial energy	3600 kJ
Acoustic bit rate	10 kbps
Optical bit rate	1 Mbps

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
