# Peer review of "A Novel Energy-Efficient Contention-Based MAC Protocol Used for OA-UWSN"

_sensors, 2019, doi:10.3390/s19010183_

Round 1
Reviewer 1 Report
@page { margin: 0.79in } p { margin-bottom: 0.1in; line-height: 120% }
The paper proposes a contention based MAC protocol applicable to hybrid underwater networks with both an acoustic and an optical physical layer on the nodes. Such a technology is often referred to as multimodal communications.
The paper contains several typos and wrong language uses. If the authors really want to make an impact with this paper, they need to improve its state of proofreading considerably.
Other comments in order of appearance in the paper:
1) The literature review scans a few comparatively old papers about MAC protocols for underwater networks. Such review should be improved with recent papers about both MAC and routing in multimodal networks. There is at least one recent paper on multimodal scheduling by Diamant et al, several works by Basagni et al, as well as, e.g., the paper about the MURAO protocol. None of them are covered here.
2) Line 97: what do you mean that mobile nodes include sink nodes? It is not clear whether your protocol can support this scenario. More details and a better explanation are expected here.
3) The paragraph from lines 104 to 110 is very misleading: you discuss a sequence of a) acoustic handshake; b) optical handshake; c) actual data transfer. You present such a sequence as if this were the only way to carry out multimodal communications, whereas this is just your design choice. Several papers in the literature (including some of those already included in your account of related work) actually implement simultaneous channel probing mechanisms and choose the most convenient option. Others optimally administer data access with routing in mind. Please improve your statement and the paragraph contents.
4) Lines 126-128: how do you know the location of a node? This is quite difficult information to obtain or estimate. Please provide the details. If the location is obtained through a specific protocol, explain how such protocol integrates with your MAC, as normally a location estimation protocol sits on top of the MAC, not aside. This is a quite important aspect for the paper to be accepted.
5) From lines 135-140 we infer that you never send data through the acoustic channel. Is this actually the case? If yes, why? Please provide the details.
6) Page 5: what you call “propagation rate” is actually the bit rate of a given channel. Please correct this issue throughout the paper.
7) Calling “a” and “b” the acoustic and optical bit rates clashes with the optical channel model, where a(\lambda) and b(\lambda) indicate the light intensity attenuation due to absorption and scattering.
8) Formula (10) cannot be used “regardless of specific propagation conditions, as you state on line 232. Equation (10) is only valid in impulsive channels with a single path, and a(f) is usually measured in deep waters precisely for this reason. Be more accurate in this explanation.
9) Table 3: the values chosen for \alpha and \beta are very low, and thus completely unrealistic. How did you get these values?
10) Table 4 reports an initial node energy of 1 W. Besides the fact that Watts are a measure of power, not energy (which is measured in Joules), a starting energy of 1 Joule is extremely low. Typical batteries provide several MegaJoules of energy to a node. Please correct both this and the values mentioned in comment 9).
11) The performance comparison in section 5.2 is not fair. Comparing your protocol against pure slotted FAMA obvious yields your protocol a clear winning case. You need to design a simple optical/acoustic version of slotted FAMA, and to compare against that version. A simple case could be a version that exchanged RTS/CTS messages over both acoustics and optics at the same time, and uses optics whenever a link is available, and acoustics otherwise.
12) Figure 8: the first comment is that a lifetime of less than 400 seconds is useless for any practical application. You need to tune the energy numbers and the traffic requirements to get realistic lifetime figures. Moreover, you improve lifetime by as little as 90 seconds: considering that integrating optics and acoustics into the same node requires very considerable hardware design efforts, the reader is left wondering if this is really a reasonable endeavor to finally achieve such a short lifetime improvement.
Author Response
Point 1: The literature review scans a few comparatively old papers about MAC protocols for underwater networks. Such review should be improved with recent papers about both MAC and routing in multimodal networks. There is at least one recent paper on multimodal scheduling by Diamant et al, several works by Basagni et al, as well as, e.g., the paper about the MURAO protocol. None of them are covered here.
Response 1: We thank this reviewer very much for the valuable comment. In the first section, we have read the articles of Professors Diamant, Basagni, etc., and added references.
Make changes in lines 54 to 57 and lines 70 to 71 in the revised version marked with red.
Point 2: Line 97: what do you mean that mobile nodes include sink nodes? It is not clear whether your protocol can support this scenario. More details and a better explanation are expected here.
Response 2: We thank this reviewer very much for the valuable comment. These sink nodes are also mobile, we describe them more carefully in the manuscript, and modify the annotations in Figure 1.
Make changes in lines 103 and 114 to 119 in the revised version marked with red.
Point 3: The paragraph from lines 104 to 110 is very misleading: you discuss a sequence of a) acoustic handshake; b) optical handshake; c) actual data transfer. You present such a sequence as if this were the only way to carry out multimodal communications, whereas this is just your design choice. Several papers in the literature (including some of those already included in your account of related work) actually implement simultaneous channel probing mechanisms and choose the most convenient option. Others optimally administer data access with routing in mind. Please improve your statement and the paragraph contents.
Response 3: We are very grateful to this reviewer for her/his precious time in reviewing our submission. We explain in the following point.
In the paper, we study the MAC protocol of the data link layer of OA-UWSN. The OA-CMAC protocol we designed includes a) acoustic handshake; b) optical handshake; c) the actual data transfer. The three parts of our design are conceived. We have not found other MAC protocols of OA-UWSN now, only routing protocol, such as MURAO routing protocol designed by Tiansi Hu and Yunsi Fei.
Thank for the reviewer’s suggestions and now we add some explanations in lines 70 to 71 marked with red.
Point 4: Lines 126-128: how do you know the location of a node? This is quite difficult information to obtain or estimate. Please provide the details. If the location is obtained through a specific protocol, explain how such protocol integrates with your MAC, as normally a location estimation protocol sits on top of the MAC, not aside. This is a quite important aspect for the paper to be accepted.
Response 4: We thank this reviewer very much for the valuable comment. Before the data transmission, the mobile node is bound to a pressure sensor, which can measure its depth and send an acoustic interrogation signal at a certain frequency. This signal can reach the water surface directly at Time0, or it can be forwarded by the submarine transponder and reach the water surface at Time1. The time difference and the measured depth are used to locate the mobile node. Then the node sends the location information through the acoustic handshake.
Make changes in lines 135 to 139 and lines 160 to 163 in the revised version marked with red.
Point 5: From lines 135-140 we infer that you never send data through the acoustic channel. Is this actually the case? If yes, why? Please provide the details.
Response 5: We thank this reviewer very much for the valuable comment. We feel sorry based on this comment that there is no more detailed description in the manuscript which lead to misunderstand. If the optical handshaking is not completed within the time slot during transmission, the node that competes to the channel through the acoustic handshake performs acoustic communication with the upper node. If the two handshaking are successful, optical data communication is performed.
We add more details in lines 149 to 151 of the MAC protocol in section 3.2.
Point 6: Page 5: what you call “propagation rate” is actually the bit rate of a given channel. Please correct this issue throughout the paper.
Response 6: We thank this reviewer very much for the valuable comment. We regret that we did not double check the writing. In order to make the paper more readable and more elaborate, we have checked the paper carefully and polished the manuscript in the revised version.
Make changes in lines 179, 188 and 283 in the revised version marked with red.
Point 7: Calling “a” and “b” the acoustic and optical bit rates clashes with the optical channel model, where a(\lambda) and b(\lambda) indicate the light intensity attenuation due to absorption and scattering.
Response 7: We thank this reviewer very much for the valuable comment. We regret that we ignored this issue. In order to make the paper more readable and more elaborate, we have checked the paper carefully and polished the manuscript in the revised version. “x” and “y”represent respectively the acoustic and optical bit rates.
Make changes in lines 179 and formula (2) (3) (4) in the revised version marked with red.
Point 8: Formula (10) cannot be used “regardless of specific propagation conditions, as you state on line 232. Equation (10) is only valid in impulsive channels with a single path, and a(f) is usually measured in deep waters precisely for this reason. Be more accurate in this explanation.
Response 8: We thank this reviewer very much for the valuable comment. We feel sorry that our description is not rigorous enough. In the manuscript, we quote Equation 10 of Reference 29 (Underwater Acoustic Communication Channels: Propagation Models and
Statistical Characterization), which is used to express large-scale path loss, considering only communication distance, not considering multipath, not channel loss modeling. In Equation 10, we have a more accurate description.
Make changes in lines 254 marked with red and figure 8 in the revised version.
Point 9: Table 3: the values chosen for \alpha and \beta are very low, and thus completely unrealistic. How did you get these values?
Response 9: We thank this reviewer very much for the valuable comment. Sorry, the energy consumption model we used in the manuscript is the sea surface energy consumption model, so the parameter values used are relatively small. In the revised version we have now replaced the energy consumption model. We refer to the article <
Make changes in lines 263 to 274 in the revised version.
Make changes in table 4 and lines 327 to 341 marked with red and figure 8 in the revised version.
Make changes in table 4 in the revised version marked with red.

Reviewer 2 Report
This paper presents a MAC protocol for hybrid underwater networks that is able to support acoustic or optical transmissions. The protocol is claimed to have low energy consumption and higher throughput than alternative proposals in the literature of multimodal communications. The authors must address some changes and clarifications in the manuscript before this being considered for publication. The following are some of my concerns.
1) It is not clear whether the topology depicted in Figure 1 (taken from its previous work [12]) is the one used in the simulations, or if it is realistic. The figure includes mobile nodes and fixed nodes, but mobility seems not to be taken into account neither in the protocol design nor in the experimental validation. Can you clarify this point?
2) The description of the protocol (Section 3.1) lacks technical detail. In its design, the MAC protocols blends ideas from classical CSMA/CD (backoff) and CSMA/CA (channel probing/reserve). What are the durations of the reserve signals RTS, acoustic and optical? What is the duration of backoff? How is the distance to the fixed nodes estimated or measured? Include this information not only for completeness, it also has an impact on performance (Section 3.3). Improve figure 3.
3) Step 4 of the MAC protocol says that only if the two handshake procedures are successful, data transmission can begin on the acoustic channel or the optical channel. This seems inefficient. If at least one of the communications modes is available, why not to use it? Also, the analysis of the protocol (Section 3.3) does not seem to capture this feature.
4) Section 3.3 on the upper limit to the optical handshake time needs better explanation. First, the optical and acoustic propagation rates (called a and b) are really transmission rates in bits per second. Correct this. Second, the handshake times c and d are taken as fixed, independent of the locations of the sender/receiver. This is not realistic, unless these parameters represent upper bounds on those delays. Please, clarify this issue too. Third, line 158 seems to contain an error when it says "the success rate of the acoustic handshake is 1, and the optical handshake success rate is 1". Looking at the formula, I guess this should say 1 - \alpha and \alpha, respectively. But data transmission only happens if the two handshakes sucked, as described in Section 3.1. Is that so?
5) Equation (11) uses \alpha in a different meaning that in equation (6).
6) In the simulations, the distance between nodes is taken as fixed (30m), according to Table 4. The issues of mobility in this protocol is not well addressed, and it is an important design point.
7) Figure 5-7 depict the performance results in comparison to O-A handshake and S-FAMA. The reasons for inclusion in the plots of S-FAMA is not clear, since it is not comparable at all with the proposed protocol. It is obvious from the description that S-FAMA performs with much lower throughput, for it only uses acoustic communications. As a baseline, it does not make much sense to choose it for comparison with your work. In Figure 8, the Y axis measures energy but uses mW as units, correct this. The gain in lifetime over OA-CMAC without threshold or S-FAMA is not substantial, and in absolute terms, under 400 s. This seems too short for practical applications, and must be discussed properly.
Finally, the paper needs significant editing of the text for correcting typos,
errors and general expression in English.
Author Response
Point 1: It is not clear whether the topology depicted in Figure 1 (taken from its previous work [12]) is the one used in the simulations, or if it is realistic. The figure includes mobile nodes and fixed nodes, but mobility seems not to be taken into account neither in the protocol design nor in the experimental validation. Can you clarify this point?
Response 1: We thank this reviewer very much for the valuable comment. The network topology in the manuscript is designed in the simulation, and if used in a real-life scenario, it is suitable for use in shallow water. In Figure 1, we have not described very clearly, and Figure 1 is now modified.
The mobility of nodes is also considered in the design of the MAC protocol. If the node moves and the position information changes, the optical alignment is re-aligned in the optical handshake phase. During the experimental verification process, node mobility is also considered. In table4, the distance between the mobile node and the fixed node is set to be evenly distributed between 20m and 40m, and the mobility of the node is considered.
Point 2: The description of the protocol (Section 3.1) lacks technical detail. In its design, the MAC protocols blends ideas from classical CSMA/CD (backoff) and CSMA/CA (channel probing/reserve). What are the durations of the reserve signals RTS, acoustic and optical? What is the duration of backoff? How is the distance to the fixed nodes estimated or measured? Include this information not only for completeness, it also has an impact on performance (Section 3.3). Improve figure 3.
Response 2: We thank this reviewer very much for the valuable comment. During the simulation process, the actual acoustic handshake time varies with the distance between nodes. In the manuscript, considering that the distance between mobile node and fixed node is set to move within the range of 20m-40m (Make changes in table 4 in the revised version with red texts.), the upper limit of the acoustic handshake time in each frame is set to 0.4S. In order to obtain the upper limit of the optical handshake time and the success rate of the optical handshake as shown in Figure 4.
In one frame, since the frame length is set to 2s. After careful consideration, the use of the backoff description in the text is not accurate enough to change it to postpone access, the postponed access time is the frame length minus the audio handshake time is 1.6s.
The fixed nodes are deployed on the seabed and are evenly distributed at intervals of 30 meters in the range of 500m*500m.
Point 3: Step 4 of the MAC protocol says that only if the two handshake procedures are successful, data transmission can begin on the acoustic channel or the optical channel. This seems inefficient. If at least one of the communications modes is available, why not to use it? Also, the analysis of the protocol (Section 3.3) does not seem to capture this feature.
Response 3: We thank this reviewer very much for the valuable comment. In the manuscript, regarding the OA-CMAC protocol we designed, if the acoustic handshake is successful, the optical handshake is unsuccessful, then the acoustic communication is performed; if the two handshakes are successful, the optical communication is performed. The reason for this design is that optical communication has the advantages of high speed and low power consumption. If optical communication can be used between nodes, it should be preferred.
In the manuscript, we are sorry that we did not make it clear. We have added more detailed descriptions in the revision.
Make changes in lines 114 to 119 in the revised version with red texts.
Point 4: Section 3.3 on the upper limit to the optical handshake time needs better explanation. First, the optical and acoustic propagation rates (called a and b) are really transmission rates in bits per second. Correct this. Second, the handshake times c and d are taken as fixed, independent of the locations of the sender/receiver. This is not realistic, unless these parameters represent upper bounds on those delays. Please, clarify this issue too. Third, line 158 seems to contain an error when it says "the success rate of the acoustic handshake is 1, and the optical handshake success rate is 1". Looking at the formula, I guess this should say 1 - \alpha and \alpha, respectively. But data transmission only happens if the two handshakes sucked, as described in Section 3.1. Is that so?
Response 4: We are very grateful to this reviewer for her/his precious time in reviewing our manuscript. We thank the reviewer for his/her comments and suggestions which help us so much in revising the paper.
We explain in the following three points.
(1) We regret that we did not double check the writing. In order to make the paper more readable and more elaborate, we have checked the paper and changed propagation rate to bit rate. All modifications are marked in lines 179, 188 and 283 with red words.
(2) c and d represent the acoustic handshake time and the optical handshake time, respectively, which vary with the distance between the nodes. In the simulation of Figure 4, in order to obtain the upper limit of the optical handshake time under the fixed acoustic handshake time, c is set to 0.4s, the 0.4s set at this time is the same as the time limit for the handshake. The time required for the actual handshake will change with the distance. Since the distance between the nodes is 20-40m, the upper limit is set to 0.4s is enough. Thereby ensuring that optical communication has better throughput than acoustic communication.
(3) In the manuscript, the success rate of the acoustic handshake and the optical handshake is written as 1 duo to mistakes. Because both of them are variables, they have been corrected on the modified version. Now the success rate of the acoustic handshake and the optical handshake is written as and in line 180.
Data transfer occurs after two handshakes, but not after the two handshakes are successful. If the acoustic and optical handshake is successful, optical communication is performed; only the acoustic handshake is successful, and the optical handshake fails to perform acoustic communication.
Point 5: Equation (11) uses \alpha in a different meaning that in equation (6).
Response 5: We thank this reviewer very much for the valuable comment. In the revision, we have changed Equation 11, which does not contain \alpha.
Point 6: In the simulations, the distance between nodes is taken as fixed (30m), according to Table 4. The issues of mobility in this protocol is not well addressed, and it is an important design point.
Response 6: We thank this reviewer very much for the valuable comment. Sorry, there is no clear statement in the manuscript. The distance between the nodes in Table 4 represents the distance between the fixed nodes is 30m. The distance between the mobile node and the fixed node is variable, and is uniformly changed from 20m to 40m in the simulation. Therefore, we consider the mobility between nodes.
Make changes in table 4 in the revised version with red texts.
Point 7: Figure 5-7 depict the performance results in comparison to O-A handshake and S-FAMA. The reasons for inclusion in the plots of S-FAMA is not clear, since it is not comparable at all with the proposed protocol. It is obvious from the description that S-FAMA performs with much lower throughput, for it only uses acoustic communications. As a baseline, it does not make much sense to choose it for comparison with your work. In Figure 8, the Y axis measures energy but uses mW as units, correct this. The gain in lifetime over OA-CMAC without threshold or S-FAMA is not substantial, and in absolute terms, under 400 s. This seems too short for practical applications, and must be discussed properly.
Response 7: We thank this reviewer very much for the valuable comment.
We explain in the following two points.
(1) In the manuscript, OA-CMAC compares with the optical-acoustic fusion MAC protocol OA-MAC based on node time slot allocation. Since the MAC protocol of other optical-acoustic fusion sensor networks other than OA-MAC is not found, the acoustic-based S-FAMA protocol is added to the simulation. The comparison is to show that the OA-CMAC protocol designed in this paper is superior to the current acoustic communication sensor network in terms of unit time throughput and node lifetime.
(2) We feel sorry to confuse power with energy, we have checked the paper carefully and polished the manuscript in the revised version. According to the actual situation, we change the initial energy of the node to 3600kJ in table 4, so the unit of the abscissa is changed from seconds to hours. Combined with Figure 5.6.7 and Figure 8, OA-CMAC not only increases the life cycle by 39%, but also has a many times higher throughput per unit time, so it is worthwhile.

Round 2
Reviewer 1 Report
@page { margin: 0.79in } p { margin-bottom: 0.1in; line-height: 120% }
I will go through the answers of the authors to my previous comments (my addition is marked by >>>)
Response 1)
>>> I would not call MURAO “the only mature protocol”, as MURAO has only been tested in simulations, whereas other protocols you now cite have actually been tested in sea experiments. Please rephrase that.
Also note that references 10 and 21 are the same.
Response 2)
>>> OK
Response 3)
>>> It is not true that there are no other MAC protocols in the literature. For example, your references 9, 16, and 36 all talk about MAC protocols. Please correct.
Response 4)
>>> Your answer implies that the nodes are synchronized. A common time reference is also quite challenging to achieve, so that you basically moved the issue of localizing a node to the issue of synchronizing it. Unfortunately the answer is unconvincing in this respect, as the synchronization of a whole network requires significant overhead, and is quite uncommon to have, making the protocol design impractical.
Response 5)
>>> OK
Responses 6) and 7)
>>> OK
Response 8)
>>> OK
Response 9)
>>> I maintain that the parameters you employ are not representative of typical underwater transceivers. P = 3 mW is realistic for low-power listening states, but for transmission you typically have power consumptions on the order of watts or tens of watts. Also, typically the power consumption is not tunable with infinite granularity, but you normally have 2-4 power levels that you could choose among. To simplify the problem, you should avoid considering per-packet perfectly tuned power consumption, and rather assume that always the same power is drawn from the battery for each transmission.
You may also consider to scale up the transmission area.
Response 10)
>>> OK
Response 11)
>>> Sorry but I do not think that your explanation dissipates my doubts about the lack of fairness in your protocol comparisons. I still believe you should implement a double optical/acoustic slotted FAMA protocol which exchanges RTS/CTS messages over both acoustics and optics at the same time, and uses optics whenever a link is available, or acoustics otherwise.
Response 12)
>>> Partly OK. See my comment about the power consumption above.
Author Response
Thank you very much for your proposed changes. We mark the first round of changes as blue and the second round, which is the part of this revision, as red.

Reviewer 2 Report
The new version of the paper incorporates some substantial changes and improvements over the first version. In particular, some of my previous concerns have been adequately addressed:
1) The MAC protocol is better explained and justified (Section 3.2)
2) Some inconsistencies in the notation and assumptions in the analysis have been fixed (Section 3.3)
3) The model of energy consumption (Section 4.2.2) is now more clear.
While these are significant steps to improve the paper, there are still some issues that the authors have not addressed properly, in my opinion.
1) First, the experimental methodology is quite simplistic and lacks detail. The simulated topology consists of a total of 8 nodes, a small number in a realistic setting. The model of mobility for the mobile nodes is not specified (it is only said that the distance ranges between 20-40m), and the average distance is not considered in the performance plots.
2) The amount of energy has been increased, so that the lifetime of the system is now realistic. However, in the plots, the results of S-FAMA are still included. S-FAMA is much simpler and based on completely different assumptions, so I wonder why to include it. Certainly, it is not a fair comparison between the protocols.
3) The analysis included in Section 3.3 has been changed slightly. Now, it includes the probabilities of success both for the acoustic and the optical handshakes. Nevertheless, it continues to be a simplistic analysis of the throughput, in that the handshake times are considered as fixed. The paper does not provide much detail about the location of the nodes (which is in itself a non-trivial task), but however this is done, the handshake time depends on the relative positions of sender and receiver. The authors must clarify whether the time in formula (2) are bounds or are considered fixed for some reason.
Regarding the presentation, I would suggest moving the figures 2,3,4, 6 & 7 to
the top of the page. The same for the tables included in the manuscript.
Author Response

(The authors gave the same response as above.)

Round 3
Reviewer 2 Report
The authors have addressed correctly my previous concerns with this manuscript. I would only suggest a thorough revision of the English language in order to improve the quality of the presentation.
Author Response
We are very grateful to this reviewer for her/his precious time in reviewing our paper. We have polished the manuscript in the revised version marked with red in 13 points.
